# Using *Rutilus rutilus* (L.) and *Perca fluviatilis* (L.) as Bioindicators of the Environmental Condition and Human Health: Lake Łańskie, Poland

**DOI:** 10.3390/ijerph17207595

**Published:** 2020-10-19

**Authors:** Joanna Łuczyńska, Beata Paszczyk, Marek Jan Łuczyński, Monika Kowalska-Góralska, Joanna Nowosad, Dariusz Kucharczyk

**Affiliations:** 1Chair of Commodity and Food Analysis, University of Warmia and Mazury in Olsztyn, ul. Plac Cieszyński 1, 10-726 Olsztyn, Poland; paszczyk@uwm.edu.pl; 2The Stanisław Sakowicz Inland Fisheries Institute in Olsztyn, ul. Oczapowskiego 10, 10-719 Olsztyn, Poland; mj.luczynski@infish.com.pl; 3Department of Limnology and Fishery, Institute of Animal Breeding, Faculty of Biology and Animal Science, Wrocław University of Environmental and Life Sciences, ul. J. Chełmońskiego 38 c, 51-630 Wrocław, Poland; monika.kowalska-goralska@upwr.edu.pl; 4Department of Ichthyology and Aquaculture, Warmia and Mazury University, Al. Warszawska 117A, 10-701 Olsztyn, Poland; nowosad.joanna@gmail.com (J.N.); darekk56@gmail.com (D.K.)

**Keywords:** freshwater fish, total mercury, BCF, THQ, lipid quality indices

## Abstract

The aim of this study was to determine the mercury content and fatty acids profile in roach (*Rutilus rutilus* L.) and European perch (*Perca fluviatilis* L.) from Lake Łańskie (Poland). Mercury content was higher in the muscles than other organs in both species (*p* < 0.05). Mercury accumulates along the food chain of the lake’s ecosystem. The value of the bioconcentration factor (BCF) indicated that Hg had accumulated in the highest amounts in muscles and in the other organs as follows: muscles > liver > gills > gonads. The metal pollution index (MPI) and target hazard quotient (THQ) were below 1, which means that these fish are safe for consumers. The values of HIS, GSI and FCF indicators show that both species of fish can be good indicators of water quality and food contamination. There were few differences between fatty acid content in the muscles of perch and roach. Contents of fatty acids having an undesirable dietary effect in humans (OFA—hypercholesterolemic fatty acids) were lower compared to hypocholesterolemic fatty acids (DFA, i.e., the desirable ones). In addition, the lipid quality indices AI and TI in the muscles of fish were at 0.40 and 0.22 (perch) and at 0.35 and 0.22 (roach), respectively. On this basis, it can be concluded that the flesh of the fish studied is beneficial from the health point of view.

## 1. Introduction

The quality of the aquatic environment and its impact on the organisms that live in it, especially those that are subject to human consumption, rise serious concerns these days. Among the lake ichthyofauna, several species best indicate the quality of the reservoir. These include vendace (*Coregonus albula* L.), smelt (*Osmerus eperlanus* L.), bleak (*Alburnus alburnus* L.), perch (*Perca fluviatilis* L.) and roach (*Rutilus rutilus* L.) [1,2]. Two processes are used to assess the quality of both the reservoir and the fish based on their contamination, namely bioconcentration and bioaccumulation. The bioconcentration index (BCF) is used to estimate the contamination in trophic chains based on the information about pollutant concentration of substances in the body and in the external environment [2,3]. It usually differs among various pollutants and even within one species [2], and can take values between 0 and infinity [4]. According to Sauliutė et al. [3], the higher the index, the more intense the bioconcentration of metals in fish. Contrary to the bioconcentration process in fish, which involves the absorption of chemicals from water through the respiratory surface and/or the skin, another process called bioaccumulation addresses all exposure routes, including food ingestion. However, like BCF, the bioaccumulation factor (BAF) in fish is the ratio of the concentration of a chemical in the organism to its concentration in water [5,6]. According to Zeitoun and Mehana [7], this index is used to assess the concentration of heavy metals in various fish tissues. As already mentioned, both of these processes take place in the body and allow the determining of the quality of water and aquatic organisms in terms of mercury contamination. Mercury is accumulated in aquatic food chains and can undergo the biomagnification process, hence it can pose a threat to human health [8]. It is known that fish, as well as water mammals and waterfowl used as food sources, are important sources of mercury in some populations, especially those who eat fish or wild game from the top of the food chain (i.e., larger fish and larger mammals) [9].

Fish for human consumption are acquired from two main sources. One is the intensive breeding combined with artificial reproduction in captivity [10,11,12,13,14] and the other is fishing in oceans, seas, lakes and rivers. Fish farming, especially in recirculated aquaculture systems (RAS), is carried out under strict control of both the physicochemical parameters of water and food quality (e.g., [15,16]). However, in the wild, fish eat food that they find, which may contain contaminants that are harmful not only to the fish themselves, for example, by interfering with reproductive efficiency (e.g., [17]), but also potentially to humans who consume these fish. This is particularly important because accumulated deposits, including heavy metals, can be actively moved across the body of fish, especially during the development of gonads and gametes before reproduction, and their concentrations may change between different fish organs [18,19]. On the other hand, fish and fish products are ideal human food, because they are the source of minerals, vitamins, proteins and fatty acids, especially the long-chain *n*-3 polyunsaturated fatty acids [20,21]. The main functions of fatty acids include: great sources of energy—high energy per gram (37 kJ/g fat), and transportable forms of energy—blood lipids; storage of energy, e.g., in adipose tissue and skeletal muscle; component of cell membranes; insulating agents—thermal, electrical and mechanical insulation; and signaling molecules—eicosanoids and gene regulation (transcription) [22]. Apart from the fact that eating fish is part of the cultural traditions of many people [23], it is also known that the diet consisting of fish meat has many benefits for the human body, because it has the lowest level of saturated fatty acids (SFA), which are responsible for the increased incidence of cardiovascular diseases [24]. One of the factors of this disease is the elevated level of LDL cholesterol fraction [25]. According to Lunn and Theobald [25], replacing SFA with monounsaturated (MUFA) or polyunsaturated fatty acids *n*-6 PUFA reduces the level of LDL (“bad”) cholesterol. According to Djazayery and Jazayery [26], α-linolenic acid (ALA), eicosapentaenoic acid (EPA) and docosapentaenoic acid (DHA), representing *n*-3 polyunsaturated fatty acids (PUFA), play important roles, especially in the treatment of cardiovascular diseases and other hazards that affect human health and life. The same opinion is shared by other authors showing both the causes and effects of the low consumption of foods rich in these fatty acids [27,28,29,30,31,32,33,34,35,36].

Roach and perch are the indicator fish of Europe’s freshwater and brackish water. They live in many types of waters and can eat very diverse food, which makes them reflect the cleanliness of the environment they inhabit (e.g., [37,38,39]). The health benefits and risks are likely to vary according to the fish species and fish size, harvesting and cultivation practices, and the frequency and amount consumed and the way in which it is served [23,40,41]. Consequently, the aim of this study was to determine:Differences between mercury content, hepatosomatic index (HSI) and gonadosomatic index (GSI) in two freshwater fish species examined.Differences between mercury content in organs (muscles, liver, gills and gonads) of the same species.The impact of biometric parameters (body weight and total length) and Fulton’s condition factor (FCF) on the content of mercury in selected organs of fish.The health risk posed by mercury using estimated daily intake (EDI), tolerable weekly intake (EWI) and target hazard quotient (THQ).Metal pollution index (MPI) and bioconcentration factor (BCF) based on the mercury content of the four organs.Differences between the fatty acids profile and the lipid quality indices (atherogenicity index (AI), thrombogenicity index (TI), flesh-lipid quality index (FLQ), hypercholesterolemic fatty acids (OFA) and hypocholesterolemic fatty acids (DFA)) in muscles of perch (*Perca fluviatilis* L.) and roach (*Rutilus rutilus* L.).

## 2. Materials and Methods

### 2.1. Sampling and Sample Preparation

Perch (*Perca fluviatilis* L.) (*n* = 9) and roach (*Rutilus rutilus* L.) (*n* = 10) were caught from Lake Łańskie, which is located in the vast Ramuckie Forests, about 15 km south of Olsztyn. The lake is one of the largest and deepest lakes in the Olsztyn Lake District (northeastern Poland) (Figure 1), located in the Protected Landscape Area of the Wilderness Napiwodzko—Ramucka, Natura Areas 2000 PLB280007—Wilderness Napiwodzko—Ramucka and PLH28005—Refuge Napiwodzko—Ramucka. The area around the northeastern shore of the lake is one of the largest forest reserves of the voivodeship, called Warmiński Forest. In the southwestern part of the catchment, farmland prevails. The lake does not receive pollutants from point sources. Its bottom is diversified, with numerous gullets and underwater hills. Tourists come to Lake Łańskie not only from Poland, but also from abroad, while fish from this lake belonging to the Fisheries Farm are marketed on the local market and beyond. According to the research carried out by the Provincial Inspectorate for Environmental Protection, the level of mercury and its compounds in water in the year 2013 was below the estimate by using the proper standard measurement or standard sample LOQ (limit of quantitation), set at 0.02 μg/L. 

After catching the fish, they were taken to the laboratory. On the same day they were weighed and measured. The body weight (± 0.01 g) and total length (± 0.1 cm) of each fish are presented in Table 1. Muscles (without skin) were dissected from the dorsal part, and together with liver, gills and gonads were stored at −30 °C in a refrigerator until analysis. Each sample, prepared from one specimen, was in duplicate. After thawing, the samples were ground and homogenized, then weighed in two parallel repetitions.

Ethical permit: fish were bought at the fish farm and were already dead. According to European and Polish Law, the research done on the tissue of commercially caught fish is free from obtaining permission from the Local Ethical Commission.

### 2.2. Element Analysis

#### 2.2.1. Mercury

The content of mercury was determined in four organs: muscles, liver, gonads and gills (duplicate samples) using Milestone DMA-80 with a detection limit (LOD) at 0.02 μg/kg. LOD is the lowest value that can be detected with the given analytical procedure and based on statistical significance. The parameters of the method are described in the previous publications [37,42]. The quality of the method was tested using the reference material: BCR CRM 422 (muscles of cod *Gadus morhua* (L.)) with a certified value of mercury. The recovery rate of Hg was 100.2% (*n* = 4).

#### 2.2.2. Fat and Fatty Acid Analysis

Approximately 1 g of duplicate samples (±0.0001 g) were dried to a constant weight at 105 °C in glass sample tubes with frits and transferred to weighed beakers. The lipids from the fish muscles (without skin) were hot-extracted in three steps (extraction, rinsing and drying) (E-816HE automatic extractor, BUCHI, Switzerland). 

The content of fat (%) was calculated according to the following formula: x = [(b − a) × 100]/c(1)
where a = weight of flask (g), b = weight of flask with extracted fat (g) and c = weight of samples (g).

For the determination of fatty acids, fat was extracted according to the Folch’s procedure [43]. Sample preparation for the analysis can be found in an earlier publication [30].

The fatty acids of methyl esters of each sample determined the following conditions: capillary column (dimension = 30 m × 0.25 μm, with a 0.32 mm internal diameter, liquid phase StabilwaxR), temperature: flame-ionization detector—250 °C, injector—230 °C, column—190 °C, carrier gas—helium with a flow rate 1.5 mL/min with a flame ionization detector (FID). The apparatus 7890A Agilent Technologies chromatograph (Waldbroon, Germany) was used for analysis, but individual fatty acids were identified by comparing the relative retention time peaks of Supelco’s standards (Supelco, Bellefonte, PA, USA).

#### 2.2.3. Indicators of Environmental Conditions and Fish Quality (FCF, HSI, GSI, MPI and BCF)

The Fulton’s condition factor (FCF) (Table 1) was calculated as described by Łuczyńska et al. [42] and Hamid et al. [44], whereas the value of the hepatosomatic index (HSI) and the gonadosomatic index (GSI) were estimated using the formulae (Table 1) shown in the publication of Łuczyńska et al. [38] and Sadekarpawar and Parikh [45]. The metal pollution index (MPI) was determined using the formulae by Łuczyńska et al. [38], Usero et al. [46,47] and Abdel-Khalek et al. [48] (Table 1), while the bioconcentration factor (BCF) was calculated using the formula proposed by Yarsan and Yipel [5] and Lau et al. [49].

#### 2.2.4. The Lipid Quality Indices

The atherogenicity index (AI) and the thrombogenicity index (TI) (AI and TI, respectively) were calculated as presented by Łuczyńska et al. [37], Ulbricht and Southgate [50], Garaffo et al. [51] and Telahigue et al. [52]. The flesh-lipid quality (FLQ) indicating the percentage correlation between EPA + DHA and the total lipids was determined using the formulae by Łuczyńska et al. [37], Abrami et al. [53] and Senso et al. [54]. The formulae used to calculate the hypercholesterolemic (OFA) and hypocholesterolemic (DFA) fatty acids are presented in the publication of Łuczyńska et al. [37].

#### 2.2.5. Human Health Risk Assessment

Similar to the target hazard quotient (THQ), the estimated daily intake of mercury (EDI) was calculated as described in earlier publications [30,34].

### 2.3. Statistical Analysis

Significant differences in the content of fatty acids and lipid quality indexes in the muscles of the fish examined were estimated using a one-way analysis of variance (ANOVA) after testing for the homogeneity of variance (Levene’s test) at a significance level of *p* ≤ 0.05. Similarly, differences in the mean content of mercury between species and organs of the same species were calculated using STATISTICA 12 software (StatSoft, Kraków, Poland). The correlation coefficients between the content of mercury in the organs of fish and their size (body weight and total length) were evaluated using STATISTICA 12 software.

## 3. Results

### 3.1. Mercury and Tools for Monitoring Fish and Environmental Conditions (BCF, MPI, FCF, HSI and GSI)

Mercury contents in the organs of perch were significantly higher than in the organs of roach (*p* ≤ 0.05) (Table 1). The muscles of fish studied contained significantly more mercury than the other organs. The content of mercury decreased in the following order: muscles > liver > gonads ≈ gills (perch) and muscles > liver > gills > gonads (roach) (Figure 2). Regardless of the species, a positive correlation was found between mercury content in the muscles and the fish weight (Table 2), with respective correlation coefficients determined for perch and roach at r = 0.785 (*p* = 0.012) (Figure 3a) and r = 0.777 (*p* = 0.008) (Figure 3b), respectively. A positive correlation was also found between levels of mercury in the muscles of perch and the length of this fish (r = 0.760, *p* = 0.017) (Figure 3c). All parameters studied were higher in perch than roach (Table 1).

### 3.2. Human Health Risk Assessment

In 2014, the annual fish consumption was 12.3 kg/per capita/year [55]. The estimated daily intake (EDI) of mercury from the 33.698 g portions of fish was 0.124 μg/kg/day (perch) and 0.048 μg/kg/day (roach) (Table 3). The weekly intake of mercury with the 235.89 g portion of perch and roach flesh accounted for 21.73% and 8.31% of the TWI (for inorganic mercury expressed as 4 µg/kg body weight), and for 66.87% and 25.58% of the TWI (for methylmercury expressed as 1.3 μg/kg body weight), respectively. The content of mercury was lower than the maximum acceptable level (0.5 mg/kg) estimated by the Commission Regulation (EC) No 629/2008 of 2 July 2008. The target hazard quotient (THQ) for mercury in the fish examined is shown in Table 3. 

### 3.3. Fatty Acids

Differences between Ʃ SFA (saturated fatty acid), MUFA (monounsaturated fatty acid) and *n*-3 PUFA (polyunsaturated fatty acid) in the muscles lipids of the fish examined were not statistically significant (*p* > 0.05) (Table 4). The muscles of roach contained more Ʃ n-6 PUFA (polyunsaturated fatty acid) than the muscle tissue of perch (*p* ≤ 0.05), but the *n*-3/*n*-6 ratio was statistically significantly higher (*p* ≤ 0.05) in the muscles of perch than in the same tissue of roach. The most abundant SFA in both studied fish was palmitic acid C16:0, whereas oleic acid (C18:1) was the major MUFA group. The differences in the content of palmitic acid between the muscles of perch and roach (20.64% and 20.63%, respectively) were not significant (*p* > 0.05). The percentage of oleic acid in the muscles of perch was significantly higher (14.48%) than in the muscle tissue of roach (11.17%) (*p* ≤ 0.05). Arachidonic acid (C20:4 *n*-6, AA) was the major *n*-6 polyunsaturated fatty acid, and its higher content was determined in the muscles of roach (10.04%) than perch (7.64%) (*p* ≤ 0.05). Docosahexaenoic acid (C22:6 *n*-3, DHA) was the major *n*-3 polyunsaturated fatty acid, and its contents determined in the muscle tissue of perch and roach (22.78 and 23.81%, respectively) did not differ significantly (*p* > 0.05)

### 3.4. Lipid Quality Indexes

A significantly higher index of atherogenicity (AI) characterized the muscles of perch (0.40) than roach (0.35) (*p* ≤ 0.05) (Table 2). There were no significant differences between the value of the thrombogenicity index (TI), flesh-lipid quality index (FLQ), hypercholesterolemic fatty acids (OFA) and hypocholesterolemic fatty acids (DFA) in the muscles of the fish examined (*p* > 0.05).

## 4. Discussion

Kareem et al. [59] found that a higher growth rate of fish and good habitat conditions are indicated by higher HSI values. In contrast, lower HSI values mean adverse environmental effects to fish health, which affects their growth. Therefore, the value of HSI provides valuable information not only about the health of fish, but also about the quality of the aquatic ecosystem. Pieterse [60] has stated that GSI is a biomarker of exposure to toxic fish and that histopathology is a useful tool to assess the degree of contamination. Similarly, Tsoumani et al. [61] have reported that FCF can be used as a good indicator of the quality of water ecosystems inhabited by fish or of the general health of fish populations. Based on the low MPI, it can be concluded that Lake Łańskie is not exposed to direct pollution. As suggested by Sauliutė et al. [3], the higher BCF values indicate more intense bioconcentration of metals in fish. The values of other indicators also show that Lake Łańskie is an aquatic reservoir unpolluted with mercury. According to Kuklina et al. [62], the muscles of perch had a higher mercury content than other fish from drinking water reservoirs (Czech Republic) (perch > pikeperch > rudd > tench > roach > bream). These results are consistent with those found in the present research (Table 1). In turn, predatory fish (salmon trout, pike and perch) were reported to contain more mercury than the benthophages (whitefish and bream) from the European Russian lakes and rivers. In addition, mercury content decreased as follows: salmon trout > pike > perch > whitefish > bream. The same authors found high mercury contents in the liver and muscles (liver ≥ muscles ≥ kidney > gills ≥ skeleton), and stated that liver and muscles may be recommended indicators of mercury pollution of basins (Montenegro [63]). Mercury contents were generally found to increase with trophic levels, because they were the highest (0.093 mg/kg) in high trophic level predatory fish, followed by middle trophic level predatory fish (0.063 mg/kg) and low trophic level fish (0.047 mg/kg), however, the differences were not significant (*p* > 0.05) [64]. In contrast, Kalisinska et al. [65] found no differences in mercury content between benthophages and phytophages, but showed mercury contents to differ significantly between predators and benthivores. Ðikanović et al. [66] found that the bioaccumulation of metals, including Hg, in organs of fish from West Morava River Basin (Serbia) could be ordered as follows: Prussian carp > northern pike > freshwater bream > European perch > chub > common nase > barbell > roach > European catfish. According to Łuczyńska et al. [67], the concentration of mercury in fish decreased as follows: ide ≈ perch ≈ flounder > rainbow trout > bream ≈ carp (in liver), perch ≈ ide > flounder > bream ≈ rainbow trout > carp (in muscles) and perch ≈ flounder ≈ ide > rainbow trout > bream ≈ carp (in gills) (*p* ≤ 0.05). The order of mercury accumulation in fish muscles found by Rakocevic et al. [68] was: eel > perch > rudd > carp > roach > bleak, but with no significant differences between the species. These authors reported no significant correlations between mercury content in the muscles of fish studied and their age and size. In turn, a positive correlation between total mercury concentration in the muscles of roach and perch from four lakes in the Olsztyn Lake District (Poland) was found by Łuczyńska et al. [69]. This is in accordance with the data of the present study.

The THQ values were below 1, hence it is known that there is no non-carcinogenic health risk for people by consuming meat obtained from the fish examined (Table 3). According Łuczyńska et al. [37], Sadekarpawar and Parikh [45] and Khemis et al. [58] THQ values in freshwater fish were also below 1. Kimáková et al. [70] showed that the maximum mercury level decreed by the Ministry of Aquaculture of the Slovak Republic and by the European Commission Regulation was exceeded in 50.52% of all fish studied. Large specimens of high trophic level pelagic and demersal species from the Central Adriatic and Tyrrhenian coasts of Italy exceeded the maximum limit appointed by the European Commission, whereas the authors detected a lower content of mercury in low trophic level demersal and pelagic-neritic fish and in young specimens belonging to high trophic level species [71]. In turn, the content of mercury in the muscle tissue of six different fish species from the Danube River (Belgrade) was below the maximum allowable level in the Republic of Serbia [66]. Olmedo et al. [72] found high content of mercury in some predatory fish, which was, however, below the regulatory maximum level set by the EC Regulation. According to Strandberg et al. [73], the consumption of perch from humic lakes exposed humans to greater risks (higher concentrations of mercury), and provided fewer benefits (lower concentrations of EPA + DHA) than consumption of perch from clear lakes. The total contents of EPA and DHA in 100 g of muscle tissue of the fish from the Vistula Lagoon were as follows: eel—2.11 g; herring—0.62 g; bream—0.33 g; roach—0.19 g; perch—0.16 g; and pikeperch—0.14 g (the recommended daily dose for healthy persons is 0.5 g) [74].

The chemical composition of fish (i.e., water, proteins, lipids, vitamins and minerals fatty acids) varies among one species and individual fish, but it also depends on age, sex, environment and season [75]. According to Vasconi et al. [76], planktivorous fish contained the lowest values of *n*-3 PUFA, but the highest amounts of MUFA (*p* ≤ 0.05). These authors also found that the carnivorous fish had the highest amounts of SFA and *n*-3 PUFA (*p* ≤ 0.05), but the lowest content of MUFA. In turn, Łuczyńska et al. [77] reported that non-predatory fish (bream, vendace and roach) contained lower amount of SFAs than the predatory fish (perch, pike and burbot) (Mazurian Great Lakes, Poland) (*p* ≤ 0.05). An opposite dependency was observed in the case of total *n*-3 PUFAs and EPAs (*p* ≤ 0.05), whereas the amounts of MUFAs, *n*-6 PUFAs, DHAs and the *n*-3/*n*-6 ratio were similar in the two groups of fish (*p* > 0.05). These results are not consistent with those observed in the perch and roach examined, being representatives of predatory and non-predatory fish, respectively (Table 2). Kainz et al. [78] studied dorsal muscles of the following species: Arctic charr (*Salvelinus alpinus* L.), pike (*Esox lucius* L.), perch (*Perca fluviatilis* L.), brown trout (*Salmo trutta* L.), roach (*Rutilus rutilus* L.) and minnow (*Phoxinus phoxinus* L.) from Lake Lunz, Austria and found that contents of *n*-3 and *n*-6 PUFA in these fish decreased with the increasing trophic position, which demonstrated that the bioaccumulation of these essential fatty acids did not increase with the increasing trophic level. In turn, Woźniak et al. [79] showed that the nutritional value (including fatty acids) of fish from lakes of northeastern Poland and fish farms was affected by species-specific traits, environmental conditions and aquaculture techniques.

AI and TI of the flesh of 13 freshwater fish (Czech Republic) ranged from 0.27 to 0.63 and from 0.20 to 0.61, respectively. These results suggest that the flesh of all examined species (except for Nile Tilapia) is of high nutritional quality and affords great benefits to human health [80]. According to De Sousa et al. [81], Nile tilapia from continental aquaculture in Paraiba State (Brazil) can be recommended for human consumption due to low AI and TI indices and hypocholesterolemic/hypercholesterolemic fatty acids ratio (H/H). Similarly, all of the farmed fish species except turbot (*Scophthalmus maximus* L.) had the recommended levels of the atherogenicity index, thrombogenicity index and hypocholesterolemic to hypercholesterolemic fatty acids ratio, whereas the highest flesh-lipid quality value was observed in the dentex (*Dentex dentex* L.) [82].

## 5. Conclusions

Human health is exposed in the case of contact with very harmful mercury. Perch, belonging to the last link in the trophic chain of the Łanskie Lake, due to the higher bioaccumulation capacity of mercury, should be a bioindicator of Łańskie Lake. Fish such as perch accumulated more mercury in their organs than the roach, which occupies the lower level. Although the level of mercury was higher in fish muscles than other organs, it did not exceed the acceptable standards and depended on fish size. Based on the indices that provide information on the quality of aquatic ecosystems, it can be concluded that Łańskie Lake is not exposed to mercury pollution, which is also confirmed by the research of the Provincial Inspectorate for Environmental Protection. A THQ level below 1.00 indicates that these fish are safe from a nutritional point of view, while low values of AI and TI indices and a high DFA indicate that the flesh of these fish is also richer in fatty acids, having the desired dietary effect for humans. The obtained research results indicate that even in the era of globalization, it is possible to find environments free from heavy metals, from which the fish obtained can be safely consumed by humans. At the same time, the examination of indicator fish such as roach and perch clearly indicate the quality of the aquatic environment in which these fish live.

## Figures and Tables

**Figure 1 ijerph-17-07595-f001:**
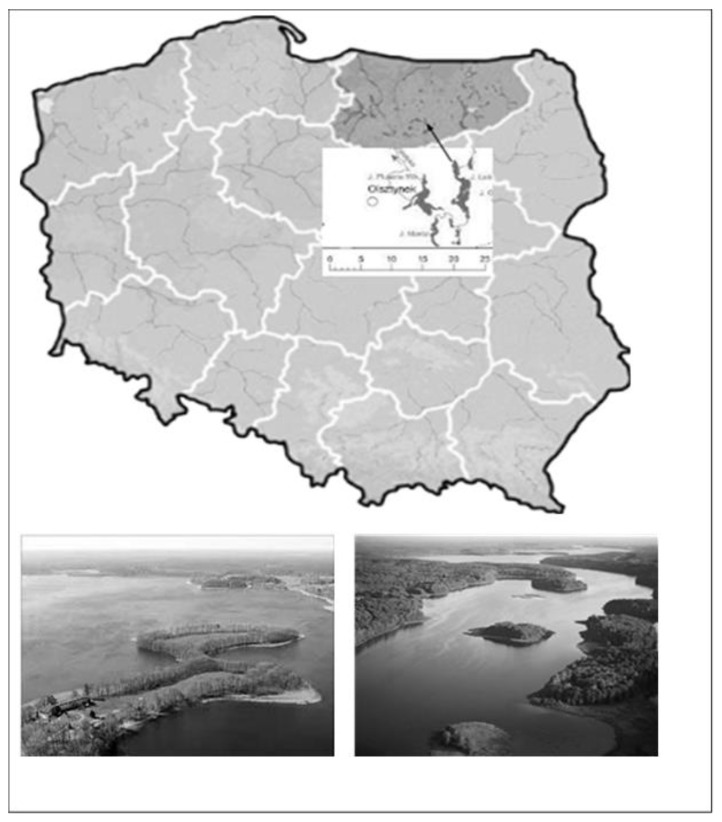
Study area was located in northeastern Poland, near the city Olsztyn (geographical coordinates: 53°58′60″ N, 20°48′08″ E).

**Figure 2 ijerph-17-07595-f002:**
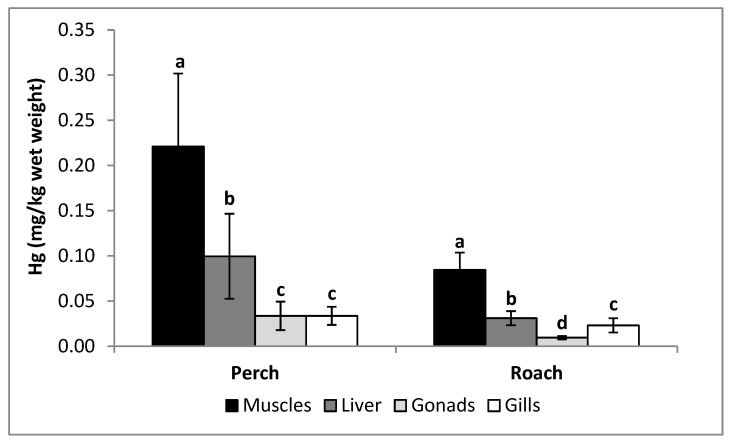
Interspecific differences (Mean ± SD) in the content of mercury in the organs of the same fish species; a, b, c and d, significant difference (*p* ≤ 0.05). The same letter indicates the absence of significant differences between organs of the same fish studied.

**Figure 3 ijerph-17-07595-f003:**
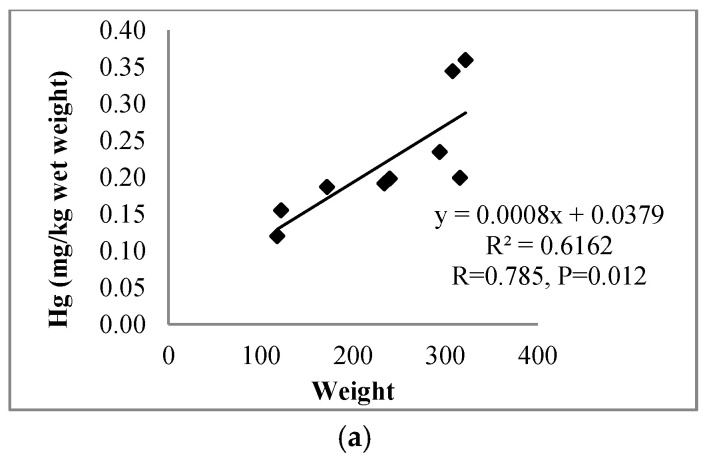
Relationship between the content of mercury in muscles and the weight and body length of the fish, (**a**) perch, (**b**) roach and (**c**) perch.

**Table 1 ijerph-17-07595-t001:** Biometric parameters, BCF and differences between the content of mercury in the same organs of fish examined (mg/kg wet weight).

	Perch (*Perca Fluviatilis* L.) (*n* = 9)	Roach (*Rutilus Rutilus* L.) (*n* = 10)
**Weight (g)**	236.22 ± 81.69	207.20 ± 38.17
**Length (cm)**	25.66 ± 2.36	26.05 ± 1.66
**FCF**	1.337 ± 0.168 a	1.165 ± 0.105 b
**HSI**	1.497 ± 0.280 a	0.957 ± 0.514 b
**GSI**	4.399 ± 6.554 a	0.963 ± 0.508 b
**MPI**	0.068 ± 0.021 a	0.027 ± 0.005 b
**Mean ± SD**		
**Muscles**	0.221 ± 0.081 a	0.085 ± 0.019 b
**Liver**	0.100 ± 0.047 a	0.031 ± 0.008 b
**Gonads**	0.034 ± 0.016 a	0.009 ± 0.002 b
**Gills**	0034 ± 0.010 a	0.023 ± 0.008 b
**BCF**		
**Muscles**	>11,056.1 ± 4036.3 a	>4229.5 ± 954.1 b
**Liver**	>4976.4 ± 2352.4 a	>1550.5 ± 389.3 b
**Gonads**	>1676.7 ± 787.5 a	>471.0 ± 96.6 b
**Gills**	>1678.9 ± 499.3 a	>1154.0 ± 393.7 b

*n*, number of fish; SD, standard deviation; Fulton’s condition factor (FCF); hepatosomatic index (HSI); gonadosomatic index (GSI); metal pollution index (MPI); bioconcentration factor (BCF); a and b, significant difference (*p* < 0.05). The same letter indicates the absence of significant differences between perch and roach.

**Table 2 ijerph-17-07595-t002:** Linear correlation coefficients (r) between mercury content in the organs of perch and roach and body weight, total length or Fulton’s condition factor (FCF).

	Weight	Length	Muscles	Liver	Gonads	Gills	FCF
Perch (*Perca fluviatilis* L.) (*n* = 9)							
Weight							

Length	0.978						
	*p* = 0.000						
Muscles	0.785	0.760					
	*p* = 0.012	*p* = 0.017					
Liver	0.484	0.545	0.408				
	*p* = 0.187	*p* = 0.129	*p* = 0.276				
Gonads	0.010	0.036	0.245	0.485			
	*p* = 0.979	*p* = 0.927	*p* = 0.526	*p* = 0.186			
Gills	−0.202	−0.140	−0.124	0.410	0.788		
	*p* = 0.602	*p* = 0.719	*p* = 0.751	*p* = 0.273	*p* = 0.012		
FCF	0.919	0.835	0.695	0.369	0.024	−0.184	
	*p* = 0.000	*p* = 0.005	*p* = 0.038	*p* = 0.329	*p* = 0.951	*p* = 0.635	
Roach (*Rutilus rutilus* L.) (*n* = 10)							
Weight							

Length	0.847						
	*p* = 0.002						
Muscles	0.777	0.448					
	*p* = 0.008	*p* = 0.194					
Liver	0.551	0.470	0.415				
	*p* = 0.099	*p* = 0.170	*p* = 0.232				
Gonads	0.497	0.620	0.443	0.567			
	*p* = 0.144	*p* = 0.056	*p* = 0.200	*p* = 0.087			
Gills	0.469	0.498	0.603	−0.149	0.404		
	*p* = 0.172	*p* = 0.143	*p* = 0.065	*p* = 0.681	*p* = 0.246		
FCF	0.291	−0.259	0.616	0.181	−0.151	−0.048	
	*p* = 0.415	*p* = 0.470	*p* = 0.058	*p* = 0.617	*p* = 0.677	*p* = 0.896	

*n*, number of fish; *p*, significance level.

**Table 3 ijerph-17-07595-t003:** The hazard quotient calculated for mercury content in the muscle tissue of the fish examined.

	EDI	TWI	%TWI *	%TWI **		THQ	References
RfD (mg/kg/day)	3 × 10^−4^						[56]
TWI (for inorganic mercury)	4						[57]
TWI (for methylmercury)	1.3						
*Perca fluviatilis* L. (*n* = 9)	0.124	0.869	21.73	66.871	0.414		This study
*Rutilus rutilus* L. (*n* = 10)	0.048	0.333	8.31	25.582	0.158		This study
*Rutilus rutilus* L. (*n* = 10)	0.040	0.280	7.00	21.50	0.135		[45]
*Perca fluviatilis* L. (*n* = 10)	0.091	0.637	15.92	49.00	0.303		[37]
*Abramis brama* L. (*n* = 6)	0.0086	0.060	1.50	4.611	0.029		
*Perca fluviatilis* L. (*n* = 5)	0.0762	0.534	13.34	41.056	0.254		
*Leuciscus idus* L. (*n* = 6)	0.0604	0.423	10.57	32.527	0.201		
*Cyprinus carpio* L. (*n* = 5)	0.0043	0.024	0.60	1.845	0.011		
*Oncorhynchus mykiss* Walb. (*n* = 6)	0.0081	0.057	1.42	4.363	0.027		
*Sander lucioperca* L. (*n* = 9)					4.97 × 10^−5^		[58]
*Cyprinus carpio* L. (*n* = 9)					1.17 × 10^−5^		

*n*, number of fish; RfD, oral reference dose (mg/kg/day); EDI is the estimated daily intake (μg/kg body weight/day); THQ, target hazard quotient; TWI = EDI × 7, tolerable weekly intake (µg/kg body weight). * TWI = tolerable weekly intake for inorganic mercury expressed as mercury (4 µg/kg body weight), ** TWI for methylmercury expressed as mercury (1.3 μg/kg body weight).

**Table 4 ijerph-17-07595-t004:** Fatty acid contents (% of total fatty acids) and index of AI and TI in the muscles lipids of the studied perch and roach (Mean ± SD).

Fatty Acid	Systematic Name	Trivial Name	Perch (*Perca fluviatilis* L.) (*n* = 9)	Roach (*Rutilus rutilus* L.) (*n* = 10)
Fat (%)			0.88 ± 0.56	0.72 ± 0.26
C12:0	dodecanoic	lauric	0.11 ± 0.02 a	0.11 ± 0.02 a
C14:0	tetradecanoic	myristic	1.94 ± 0.55 a	1.11 ± 0.37 b
C16:0	hexadecanoic	palmitic	20.64 ± 0.70 a	20.63 ± 1.43 a
C18:0	octadecanoic	stearic	5.21 ± 0.61 b	5.79 ± 0.54 a
C18:1	octadecenoic	oleic	14.48 ± 1.44 a	11.17 ± 2.78 b
C18:2(*n*-6) LA	*cis, cis*-9,12-octadecadienoic	linoleic	2.95 ± 0.57 a	2.35 ± 0.79 a
C20:4(*n*-6) AA	all *cis*-5,8,11,14-eicosatetraenoic	arachidonic	7.64 ± 1.16 b	10.04 ± 1.65 a
C18:3(*n*-3) ALA	all *cis*-9,12,15-octadecatrienoic	α-linolenic	1.93 ± 0.67 a	1.11 ± 0.68 b
C20:5(*n*-3) EPA	all *cis*-5,8,11,14,17-eicosapentaenoic	eicosapentaenoic	7.11 ± 0.46 a	6.11 ± 0.55 b
C22:5(*n*-3) DPA	all *cis*-7,10,13,16,19-docosapentaenoic	docosapentaenoic	2.72 ± 0.34 a	2.83 ± 0.21 a
C22:6(*n*-3) DHA	all *cis*-4,7,10,13,16,19-docosahexaenoic	docosahexaenoic	22.78 ± 4.4 a	23.81 ± 4.27 a
Ʃ SFA			28.93 ± 0.91 a	28.72 ± 1.56 a
Ʃ MUFA			21.68 ± 3.38 a	17.90 ± 5.28 a
Ʃ *n*-6 PUFA			13.08 ± 1.47 b	17.95 ± 1.85 a
Ʃ *n*-3 PUFA			36.31 ± 3.86 a	35.43 ± 3.35 a
Ʃ PUFA			49.39 ± 3.30 b	53.38 ± 4.36 a
Ʃ UFA			71.07 ± 0.91 a	71.28 ± 1.56 a
Ʃ *n*-3 HUFA			33.64 ± 4.46 a	34.05 ± 4.03 a
*n*-3/*n*-6			2.82 ± 0.48 a	1.99 ± 0.24 b
AI			0.40 ± 1.70 a	0.35 ± 0.02 b
TI			0.22 ± 0.02 a	0.22 ± 0.02 a
FLQ			29.89 ± 4.25 a	29.92 ± 4.14 a
OFA			22.68.41 a	21.86 ± 1.82 a
DFA			76.27 ± 0.50 a	77.07 ± 1.27 a

*n*, number of fish; SD, standard Deviation; a and b, significant difference (*p* ≤ 0.05). The same letter indicates the absence of significant differences between perch and roach. Ʃ SFA (saturated fatty acid) contains C12:0, C14:0, C15:0, C16:0, C17:0, C18:0 and C20:0; Ʃ MUFA (monounsaturated fatty acid) contains C14:1, C16:1, C17:1, C18:1, C20:1(n-7), C20:1(n-9) and C20:1(n-11); Ʃ n-6 PUFA (polyunsaturated fatty acid) contains C18:2, C18:3γ-lin, C20:2, C20:3, C20:4 and C22:5; Ʃ n-3 PUFA (polyunsaturated fatty acid) contains C18:3, C18:4, C20:3, C20:4, C20:5 EPA, C22:5 DPA and C22:6 DHA; Ʃ n-3 HUFA (highly unsaturated fatty acid) contains C20:3, C20:4, C20:5 EPA, C22:5 DPA and C22:6 DHA; AI, index of atherogenicity; TI, index of thrombogenicity; FLQ, flesh-lipid quality; OFA, hypercholesterolemic fatty acids; DFA, hypocholesterolemic fatty acids.

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
