# Peer review of "Using Rutilus rutilus (L.) and Perca fluviatilis (L.) as Bioindicators of the Environmental Condition and Human Health: Lake Łańskie, Poland"

_ijerph, 2020, doi:10.3390/ijerph17207595_

Round 1
Reviewer 1 Report
The manuscript titled “Using Rutilus rutilus (L.) and Perca fluviatilis (L.) as bioindicators of the environmental condition and human health: Lake Łańskie, Poland” provides detailed information on mercury accumulation in different tissues of perch and roach. The results show that Lake Łańskie is not exposed to mercury pollution as the mercury levels in the muscle and other tissues were below the unacceptable levels. Overall this is an interesting study and provides fundamental knowledge on mercury levels and their accumulation in fishes.
The authors performed a detailed study, measuring several factors including metal pollution, lipid quantities, etc. However, the article lacks flow in the introduction section, lacks some important information on experimental replicates.
I think issues with presentation of major results can be further resolved by providing additional details/context to set the stage for the reader so that when they get to the results and discussion sections, they make more sense. The introduction section (especially the first paragraph) lacks the flow and has several errors in sentence formation and grammar.
The methods section needs to mention total number of experimental replicates. Did the authors use n=9 and n=10 and did not have any technical and biological replications?
The conclusions section also needs improvement. It would be nice to add one more last sentence to explain how the results will be useful and what are the broader implications of this study.
There are several sentences that started with a reference (example lines 43, 239, 300) or state a reference as according to (Reference). I would suggest the authors to elaborate and use the organism name or some other detail relevant to the sentence (example – line 43 says higher the ratio, but the author fail to mention what ratio). Just providing a reference number is vague and not explanatory.
Some minor comments to be considered are mentioned below:
- Line 26 – Italicize e.
- Line 37 – replace There with These
- Lines 38-39 – do not make sense. Rephrase the sentence
- Line 39- What is ‘it’ in it is defined?
- Line 42 – What do you mean by ‘this’ in this indicator
- Line 42 - What is ‘its’ mean. Please elaborate to use the right term.
- Lines 44 - 47 – Rephrase the sentence and provide reference
- Line 60 – because they are the source
- Line 66 – People not ‘peoples’
- Line 69 – according to (ref), replacing SFA
- Line 70 - according to
- Lines 76 – 78 - Rephrase the sentence
- Line 78 – objectives of this study
- Lines 80 – 93 – capitalize the first word of each sentence and end the sentence with a full stop and not a comma
- Line 107 – insert comma after protection
- Line 111 – each sample was prepared from one specimen. Explain how the sample was prepared after it was removed from the refrigerator.
- Line 116 – Delete ‘.’ Before 2.2.1
- Line 119 – previous publications
- Line 120 – italicize the scientific name
- Line 130 – ‘was’ extracted
- Line 132 – ‘determined’ the following
- Line 138 – provide reference
- Line 139 – Rename the section instead of abbreviations. May be ‘assessing environmental and other conditions’?
- Line 144 and 153 – explain ‘pattern’ briefly
- Line 159 - provide reference
- Line 164 – similar to above, delete.
- Figures 2, 3 - Edit the scale units and have only two decimal places after the point. Use patterns for different treatments. The shades of grey are close and look similar. Capitalize the first letter of each word (similar to other tables and figures) Be consistent! The figure 3, details overlap; remove decimal places to avoid zeros on both x and y-axes. This also gives you space and better representation. Why is body length of roach missing?
- Tables 2, 3, 4 – Why some parts of the tables are colored?
- Line 233 – delete ‘.’ After 3.4
- Line 316 – should not be italicized right?
- The scientific names of organisms are not italicized in numerous places of the references section (Reference nos. 35, 36, 37, 40, 41, 43, 47, 49, 50, 62). Please double-check them!
Author Response
Response to Reviewer 1 Comments
Response 1: The authors performed a detailed study, measuring several factors including metal pollution, lipid quantities, etc. However, the article lacks flow in the introduction section, lacks some important information on experimental replicates.
I think issues with presentation of major results can be further resolved by providing additional details/context to set the stage for the reader so that when they get to the results and discussion sections, they make more sense. The introduction section (especially the first paragraph) lacks the flow and has several errors in sentence formation and grammar.
The methods section needs to mention total number of experimental replicates. Did the authors use n=9 and n=10 and did not have any technical and biological replications?
These information were corrected according to Reviewer suggestion.
Response 2: The conclusions section also needs improvement. It would be nice to add one more last sentence to explain how the results will be useful and what are the broader implications of this study.
This section was corrected according to Reviewer suggestion.
Response 3: There are several sentences that started with a reference (example lines 43, 239, 300) or state a reference as according to (Reference). I would suggest the authors to elaborate and use the organism name or some other detail relevant to the sentence (example – line 43 says higher the ratio, but the author fail to mention what ratio). Just providing a reference number is vague and not explanatory.
According to the Reviewer, the errors have been corrected.
Response 4: Some minor comments to be considered are mentioned below:
- Line 26 – Italicize e.
- Line 37 – replace There with These
- Lines 38-39 – do not make sense. Rephrase the sentence
- Line 39- What is ‘it’ in it is defined?
- Line 42 – What do you mean by ‘this’ in this indicator
- Line 42 - What is ‘its’ mean. Please elaborate to use the right term.
- Lines 44 - 47 – Rephrase the sentence and provide reference
- Line 60 – because they are the source
- Line 66 – People not ‘peoples’
- Line 69 – according to (ref), replacing SFA
- Line 70 - according to
- Lines 76 – 78 - Rephrase the sentence
- Line 78 – objectives of this study
- Lines 80 – 93 – capitalize the first word of each sentence and end the sentence with a full stop and not a comma
- Line 107 – insert comma after protection
- Line 111 – each sample was prepared from one specimen. Explain how the sample was prepared after it was removed from the refrigerator.
- Line 116 – Delete ‘.’ Before 2.2.1
- Line 119 – previous publications
- Line 120 – italicize the scientific name
- Line 130 – ‘was’ extracted
- Line 132 – ‘determined’ the following
- Line 138 – provide reference
- Line 139 – Rename the section instead of abbreviations. May be ‘assessing environmental and other conditions’?
- Line 144 and 153 – explain ‘pattern’ briefly
- Line 159 - provide reference
- Line 164 – similar to above, delete.
- Figures 2, 3 - Edit the scale units and have only two decimal places after the point. Use patterns for different treatments. The shades of grey are close and look similar. Capitalize the first letter of each word (similar to other tables and figures) Be consistent! The figure 3, details overlap; remove decimal places to avoid zeros on both x and y-axes. This also gives you space and better representation. Why is body length of roach missing?
- Tables 2, 3, 4 – Why some parts of the tables are colored?
- Line 233 – delete ‘.’ After 3.4
- Line 316 – should not be italicized right?
- The scientific names of organisms are not italicized in numerous places of the references section (Reference nos. 35, 36, 37, 40, 41, 43, 47, 49, 50, 62). Please double-check them!
These information was corrected according to Reviewer suggestion.
Reviewer 2 Report
1 The English of manuscript should be improved to eliminate some spelling and grammar mistakes.
2 Please indicate how many repetitions in the experiments.
3 More details about the experimental conditions should be provided.
4 Please explain why Rutilus rutilus and Perca fluviatilis were selected in this work.
5 Conclusions should be rearranged.
Author Response
Response to Reviewer 2 Comments
Response 1: 1 The English of manuscript should be improved to eliminate some spelling and grammar mistakes.
As suggested by the Reviewer, the publication has undergone a linguistic correction.
Response 2: 2 Please indicate how many repetitions in the experiments.
These information were corrected according to Reviewer suggestion.
Response 3: 3 More details about the experimental conditions should be provided.
As suggested by the Reviewer, the experimental conditions are listed in the "Material and methodology" section.
Response 4: 4 Please explain why Rutilus rutilus and Perca fluviatilis were selected in this work.
The information was added into the text (with References cited) according to Reviewer suggestion.
Response 5: 5 Conclusions should be rearranged.
As suggested by the Reviewer, the Conclusions have been sorted.
Reviewer 3 Report
Please rewrite the abstract. There are multiple grammatical errors. Please write the full form of OFA and DFA in line 26.
Correct these mistakes: Line 20 *indicates, lines 23, 27, 28 *fishes, line 37 *They.
Rephrase the sentence in lines 76, 77, 78. Make smaller sentences rather than a confusing and a big one.
In the Introduction section, the authors are advised to include more recent references and findings on the topic and also present a comparison between them.
The quality and legibility of Figure 1 must be enhanced.
The research design and the results must be explained clearly and in a lucid manner.
Please present the limitations and disadvantages of the proposed study.
There are numerous grammatical errors and typos throughout the manuscript. The English language and grammatical mistakes must be thoroughly revised, perhaps with help from a native English speaker.
Author Response
Response to Reviewer 3 Comments
Response 1: Please rewrite the abstract. There are multiple grammatical errors. Please write the full form of OFA and DFA in line 26.
As suggested by the Reviewer, the errors have been corrected.
Response 2: Correct these mistakes: Line 20 *indicates, lines 23, 27, 28 *fishes, line 37 *They.
As suggested by the Reviewer, the errors have been corrected.
Response 3: Rephrase the sentence in lines 76, 77, 78. Make smaller sentences rather than a confusing and a big one.
As suggested by the Reviewer, the errors have been corrected.
Response 4: In the Introduction section, the authors are advised to include more recent references and findings on the topic and also present a comparison between them.
As suggested by the Reviewer, the latest reports have been introduced in the "introduction" section.
Response 5: The quality and legibility of Figure 1 must be enhanced.
As suggested by the Reviewer, Figure 1 has been changed.
Response 6: The research design and the results must be explained clearly and in a lucid manner
As suggested by the Reviewer, the material and methodology were supplemented with the necessary information.
Response 7: Please present the limitations and disadvantages of the proposed study.
Considering the size of the samples, there is always a concern that the sample could be larger and that we may not have taken medium-sized fish, which could then result in erroneous results. However, considering the literature, the variations between the different species were similar, which suggests that we did not make this mistake.
Another limitation is a clean environment. On the one hand, we should be glad that this is the case, on the other hand, in a more polluted one, other tendencies could be observed.
These two things are basic limitations to me and can be seen as flaws.
Response 8: There are numerous grammatical errors and typos throughout the manuscript. The English language and grammatical mistakes must be thoroughly revised, perhaps with help from a native English speaker.
As suggested by the Reviewer, the publication has undergone a linguistic correction.
Round 2
Reviewer 1 Report
The manuscript provides information on mercury accumulation in different tissues of perch and roach in Lake Łańskie.
The rebuttal submitted by authors have significant changes with several additional details. The introduction and methods sections have improved. However, some of the changes to specific comments cannot be found in the manuscript.
For example, “The methods section needs to mention total number of experimental replicates. Did the authors use n=9 and n=10 and did not have any technical and biological replications?” The authors said the information was corrected, but I did not see any related changes in the rebuttal.
Some minor spacing errors still need to be considered. Example, line 345.
Author Response
Comments and Suggestions for Authors
The manuscript provides information on mercury accumulation in different tissues of perch and roach in Lake Łańskie.
The rebuttal submitted by authors have significant changes with several additional details. The introduction and methods sections have improved. However, some of the changes to specific comments cannot be found in the manuscript.
For example, “The methods section needs to mention total number of experimental replicates. Did the authors use n=9 and n=10 and did not have any technical and biological replications?” The authors said the information was corrected, but I did not see any related changes in the rebuttal.
The number of repetition were in text, e.g. Hg measurements: duplicate samples (136, 137) from each fish ; fat and fatty acids analysis were measured in duplicate from each fish (line 143). The number n=9 and n = 10 it was the number of roach and perch, respectively, taken for this study.
Some minor spacing errors still need to be considered. Example, line 345.
Text has been checked very carefully and typos and word sticking have been corrected. Text has also been grammatically checked.
The colors in Figure 2 have been changed as suggested by the Reviewer
Figure 3 shows only significant correlations, i.e. P <0.05, and the correlation between mercury content in roach muscles and body length was P = 0.194, therefore this correlation is not shown in the Figure.

Reviewer 2 Report
The manuscript's quality has been substantially improved. I recommend its acceptance for publication in its present form.
Author Response
Thank you for accepting the publication and for previous comments
Reviewer 3 Report
There are still some grammatical errors and punctuation mistakes. After this revision, it will be deemed to be published.
Author Response
Comments and Suggestions for Authors
There are still some grammatical errors and punctuation mistakes. After this revision, it will be deemed to be published.
Text has been checked very carefully and typos and word sticking have been corrected. Text has also been grammatically checked.
